# Objectively Measured Physical Activity Is Associated with Static Balance in Young Adults

**DOI:** 10.3390/ijerph182010787

**Published:** 2021-10-14

**Authors:** Wenfei Zhu, Yunfeng Li, Bingqi Wang, Chenxi Zhao, Tongzhou Wu, Tao Liu, Fangjun Sun

**Affiliations:** School of Physical Education, Shaanxi Normal University, Xi’an 710119, China; wzhu@snnu.edu.cn (W.Z.); yunfenglee@snnu.edu.cn (Y.L.); 190701@snnu.edu.cn (B.W.); zhaochenxi95@snnu.edu.cn (C.Z.); shanyaning@snnu.edu.cn (T.W.); taoliu@snnu.edu.cn (T.L.)

**Keywords:** physical activity, static balance, accelerometry

## Abstract

Purpose: Regular physical activity (PA) strengthens muscles and improves balance and coordination of human body. The aim of this study was to examine whether objectively measured physical activity (PA) and sedentary behaviors were related to static balance in young men and women. Design and setting: Cross-sectional community study. Participants: 86 healthy adults (50% women) aged 21.26 ± 2.11 years. Method: PA variables, including moderate-to-vigorous PA (MVPA), light PA (LPA), sedentary time (SED), and sedentary breaks, were measured by accelerometers on wrist (ActiGraph WGT3X-BT). The static balance was tested in the bipedal stance with eyes open or closed. The movement of the center of pressure, including total sway path length (SP), sway velocity (SV), and sway area (SA), was recorded with a three-dimensional force platform (Kistler 9287CA). The associations between PA (MVPA/LPA/SED/sedentary breaks) and static balance (SP/SV/SA) were analyzed using mixed linear regression analyses, with adjustments for condition (eyes open/closed), sex, age, body mass index (BMI), total device wearing time, and PA*condition. Data were analyzed with SPSS 24.0. Results: Better performance was observed in eyes-open condition (*p* < 0.05). MVPA was negatively associated with SA (*p* = 0.030), and SED was positively associated with SA after adjustments, respectively (*p* = 0.0004). No significance was found in the association of light PA, SED, or sedentary breaks with other static balance variables, respectively (*p* > 0.05). Conclusion: Increasing MVPA and less SED are associated with lower sway area measured by force platform, indicating more PA may related to better static balance in young adults.

## 1. Introduction

Balance is essential to ensuring safe activities of daily living for individuals, as well as the performance of safe locomotion [1]. Static balance is the ability to maintain postural stability and orientation with center of mass over the base of support and body at rest [2]. Dynamic balance is the ability to maintain postural stability and orientation with center of mass over the base of support while the body parts are in motion, which is also closely related to static balance [2]. Balance may be influenced by varied individual factors. For example, sex was one of the important factors that influenced postural balance [3]. Women showed higher postural stability in different conditions and measurements [4]. Visual inputs could also be crucial for postural control in healthy adults. Stability with the eyes closed was worse than those obtained with the eyes open [5]. The evaluation of static balance and its related factors is fundamental to determining predictors of performance, preventing lower extremity musculoskeletal injuries, and improving the efficacy of physical training and rehabilitation techniques.

Regular exercise and physical activity (PA) strengthen the muscles and improve balance and coordination, leading to fewer falls in the elderly [6,7]. PA can stimulate the sensory-motor response through reflex responses, as well as neuromuscular and biomechanical strategies of feedback and feedforward for postural adjustments [8]. Some existing observational and experimental research has shown a positive relationship between PA and balance. Daily PA has been proven to be positively associated with balance performance in older adults both in the short-term and long-term [6,8,9]. Walking can help to improve postural stability, especially static balance, in the older population [8,10]. Accelerometer-measured data showed older women who engaged in higher levels of MVPA and accumulated less SED had a significantly lower risk of falling [11,12]. It has also been shown that Tai-Chi and Pilates exercise were effective in improving balance ability [13,14,15]. On the other hand, intense PA may also cause musculoskeletal or articular injuries, which may indirectly have a negative impact on posture control and balance [16]. Thus, it remains unknown how daily PA is exactly correlated to static balance. Additionally, most current studies focused on older adults, but physical inactivity may also lead to poorer performance in balance and higher risk of fall at an earlier age. An investigation of the association between PA and static balance in younger adults who also develop an inactive lifestyle is warranted to understand to what extent PA relates to posture control and how to optimize the prescription of PA strategies or interventions designed to improve posture stability in daily activities.

In previous studies, PA was often measured subjectively by questionnaire and suffered from reporting bias [17,18]. Objective methods, such as accelerometers, offer a solution to these problems and can give objective estimates of the frequency, duration, and intensity of PA [17]. One study [19] using accelerometers reported there was an association between reductions in PA and impairments in balance among COPD patients. Few studies have been done to explore the relationship between objectively measured PA and static balance in general population. Additionally, it is still unknown whether sedentary time (SED), which is an independent risk factor, is related to balance performance. Further study is needed to investigate the association between objectively measured sedentary behaviors and balance performance.

In sum, the main aim of this study was to examine the association between objectively measured PA/sedentary behaviors and static balance. Our study could lead to the earlier identification of the sedentary population, who would benefit from primary prevention to reduce future falls. Efforts to promote assessment and education in this area will lead to a more active and healthier lifestyle in early life and generate tailored exercise interventions to reduce future decline in physical function and balance disorders.

## 2. Materials and Methods

### 2.1. Participant

Eighty-six healthy adults (50% women), aged 18–26 years, were recruited for the current study. The mean ± SD for age, body mass, and height were 21.26 ± 2.11 years, 67.28 ± 12.95 kg, and 172.95 ± 9.41 cm, respectively. The participants did not suffer from any disorders of the nervous system (e.g., Parkinson’s disease, stroke) that could have caused body balance problems or orthopaedic disorders making it impossible to maintain a standing position. Participants had no cognitive impairments and no history of vertigo or falls in the previous month. Participants taking medicaments that may cause balance disorders, and participants with significant visual impairment, were excluded. The study procedure is shown in Figure 1. The purpose and objectives of the study were explained to each participant before the study began, and written informed consent was obtained. The study has been approved by the Shaanxi Normal University Ethics Committee (202116003).

### 2.2. Objectively Measured PA

Each participant’s PA and sedentary behaviors, including light PA (LPA), moderate-to-vigorous PA (MVPA), SED, and sedentary breaks, were objectively measured by accelerometer-based activity monitors (ActiGraph WGT3X-BT, Pensacola, FL, USA). Participants were asked to wear the accelerometers on their non-dominant wrist and wear the accelerometer for seven consecutive days during all waking hours before the static balance ability tests. To improve the adherence of the device wearing, the attachment on wrists was chosen in this study. Studies reported reasonable precise estimations of PA when using wrist-worn devices [14,15]. Participants were instructed to remove devices when they were swimming, taking a bath, or any other times when they felt they were at risk of breaking or losing their accelerometers. The devices were initialized with a sampling frequency of 60 Hz. Participants included provided usable data with the criterion of >4 days with >10 h/day of wear time. Non-wear periods were defined as a string of zero counts per minute (cpm) for >60 consecutive minutes. Activity count cut-points were applied to differentiate SED (0–99 cpm), LPA (100–1951 cpm), and MVPA (>1952 cpm), respectively [20,21]. A sedentary bout was defined as consecutive minutes in which the accelerometer recorded less than 100 counts per minute. A sedentary break was defined as at least 1 min in which counts registered at least 100 counts after a sedentary bout [21]. The time spent in LPA, MVPA, and SED was expressed in min/day, and sedentary breaks were expressed in counts/hr. Data were downloaded from the monitor to a computer after completion of all activities. All the procedures mentioned above have been validated in previous studies [20,21].

### 2.3. Static Balance

The static balance data were obtained with a three-dimensional force platform (Kistler 9287CA, Winterthur, Switzerland). The force platform was fixed on the flat floor, and static balance was tested in the bipedal stance in two different conditions as below:(a)Standing on the force platform with eyes-open task,(b)Standing on the force platform with eyes-closed task.

Participants in our study were required to wear comfortable and standardized running shoes [22,23]. They were asked to remain in the bipedal standing posture on the platform with their arms down on each side of their hips and breathing normally (Figure 2). Running shoes were used to test their posture control ability in real life setting. In the eyes-open condition, the participants were asked to stand on the platform and watch the reference point, which was located on the wall 2 m away for 20 s. In the eyes-closed condition, the participants were required to stay on the force platform for 20 s with closed eyes.

Each test was performed 3 times with a recovery interval of 30 to 60 s to prevent fatigue. The movement track of the center of pressure (COP) was recorded (Figure 3). The “Measurement, Analysis and Reporting Software” (MARS, version 2.1.0.8, S2P, science to practice, Ltd., Ljubljana, Slovenia) was used for comprehensive analyses of force plate measurements [24]. There were three options in the software for footwear, including barefoot, running shoes, and high heels; the option of “running shoes” was chosen. Sway path (SP) total length (mm) refers to the common length of the trajectory of the COP sway calculated as a sum of the point-to-point Euclidian distances. Sway velocity (SV) total (mm/s) refers to the common length of the trajectory of the COP sway calculated as a sum of the point-to-point Euclidian distances divided by the measurement time. Sway area (SA) total (mm^2^) is defined as the total area swayed by the COP trajectory with respect to the central stance point, calculated as the ellipse containing 95% of COP.

### 2.4. Data Analysis

Data were analyzed with Statistical Package for Social Sciences software (SPSS Version 24.0, IBM Corporation, Armonk, NY, America). Descriptive statistics (including mean ± standard deviation) were calculated. Normality and homogeneity of variance were tested (*p* > 0.05). The difference between sex groups was tested by independent T-test. Difference between different tasks (open/closed eyes) within group (same participant) was tested by paired T-test. The association between PA and static balance under eye open/closed condition was analyzed using linear mixed-effects models. Fixed effects of PA variables (MVPA, LPA, SED, or sedentary breaks), condition (eyes open/closed), age, sex, body mass index (BMI), and the interaction between total device wearing time and the PA*condition were included in the models, and random intercepts over time were included to account for within-subject correlations. The level of significance was set to α = 0.05.

## 3. Results

Demographic characteristics of the 86 participants included in data analysis are displayed in Table 1. Age, height, and weight of men were significantly higher than those of women (*p* < 0.05), respectively. The device wearing time, SED, LPA, and sedentary breaks of women were significantly higher than those of men (*p* < 0.05), respectively. No significant difference was found in other aspects (*p* > 0.05).

The descriptive statistics of sway param are shown in Table 2. There were significantly less SPSV in the women group than in the men group (*p* < 0.05), respectively. SP, SV, and SA were all significantly less in the eyes-open condition than those in the eyes-closed condition (*p* < 0.05), respectively. No significance existed in other aspect (*p* > 0.05).

The associations between PA and static balance are presented in Table 3. MVPA was negatively associated with SA (*p* = 0.030), and SED was positively associated SA (*p* = 0.0004) in the eyes-closed condition. No associations were found in other aspect (*p* > 0.05).

## 4. Discussion

This study was one of the first to examine the relationship between objectively measured PA and static balance performance in young adults. The study indicated MVPA and SED were significantly associated with SA, respectively. More MVPA or less SED was associated with better performance in static balance. We also found that static balance was influenced by the factors of sex and visual contribution. Women, in the condition of open eyes, had better performance in static balance. Our results suggest daily PA is related to static balance and may be related to primary prevention for future falls by leading an active and healthy lifestyle. Sedentary population needs to have a static balance assessment to evaluate its future balance ability.

Participants in this study maintained a more stable position with their eyes open. The values of those static balance parameters were significantly lower in SP, SV, and SA, in the eyes-open condition, than those with in the eyes-closed condition. This is consistent with previous research. Visual, vestibular, and somatosensory systems are the main sensory systems involved in postural control and balance [25]. Visual inputs are crucial for postural control, and posture stability with the eyes closed was worse than that with the eyes open [4,7,26]. For example, Perrin et al. [27] examined the effects of physical activities on balance control in elderly people and found the stability with eyes closed was worse than those obtained with eyes open in all groups. Prado et al. [26], who investigated postural sway during dual tasks in young and elderly adults, found both age groups presented significantly larger postural sway in the mediolateral direction during the eyes-closed condition as compared with the eyes-open condition. Similar results were also reported by one recent study [28], indicating standing balance with eyes closed presented significantly larger postural sway and higher COP velocities. Thus, the influence of visual inputs should be considered in balance research because it has significant impact on objectively measured posture sway.

In this study, we observed MVPA was positively associated with the SA, and participants who accumulated more daily MVPA performed better in static balance than those with less MVPA. Several studies suggest that PA may play a role in maintaining proper postural stability. For example, one study reported older adults who had regular walking habit had significantly better postural stability under static condition and higher values of ankle plantar flexor and knee extensor strengths [10]. McMullan et al. [6] reported free-living PA improves balance performance in older healthy adults, both in the short-term and long-term. Cooper et al. [9] also found accumulated PA across life were positively associated with standing balance and suggest that promotion of leisure-time PA across adulthood would have beneficial effects on physical performance later in life. More LPA or MVPA was associated with better functional balance [29]. Using accelerometer measured data, the Women’s Health Initiative Objective Physical Activity and Cardiovascular Health study showed older women who engaged in higher levels of MVPA and accumulated less had a significantly lower risk of falling [11,12]. PA may stimulate muscular strength and endurance of the lower and upper limbs, which can contribute to balance gains. Additionally, it is possible that daily PA promoted neurophysiologic adaptations to increase reflex responses and proprioception, which in turn impact the balance. Furthermore, static balance is controlled by the ankle plantar/dorsiflexion [30,31], and PA can improve balance ability and muscle strength around ankles [32,33].

We also found less SED was associated with less SA and less sedentary lifestyle may be related to better performance in static balance in young adults. As we know, SED is an independent risk factor associated with less muscle mass and vascularization [34], and higher risk of osteoarthritis [35]. Research showed less SED could help reduce fall risk as well as improving overall health status [36]. Participants who stood up more often during sedentary periodshad more strength in lower-extremity and improved physical function [37]. Sedentary lifestyle with an insufficient number of sedentary breaks and PA predisposing to muscle weakness may have an adverse influence on the postural control system [35]. Future studies are in need to determine whether interventions promoting frequent breaks from sitting could be protective for balance.

In this study, the significance only existed in the relationship between MVPA/SED and SA. The reason may be that SA provides a more comprehensive evaluation of in all directions [24] because it is defined as the total area swayed by the COP trajectory with respect to the central stance point. In previous study, balance measures varied widely in their nature and sophistication. Sway characteristics, including length, velocity, and magnitude, are among the most common measures of balance and stability. Large and jerky amounts of sway are considered to reflect poor postural control. Research has revealed that physical activity or exercise are related to static balance objectively measured by sway on computerized platforms [38]. These sway measures have been proven to be associated with risk of falls or future injuries and impairments [7]. No significance was found in the association between PA and SP and SV in this study. The reason may be the relationship between PA and static balance was not very strong, and there are some other confounders, such as muscle strength, vestibular control, and joint health. Additionally, the results of this study were mainly focused on young adults. Research has indicated significant difference in balance and posture control in young and older adults. Better posture control ability and less sway existed in balance measures among young participants, compared to older adults [26,38]. Older adults had higher electrophysiological costs for a given balance task and so can be considered less efficient from a neuromuscular viewpoint [39]. There may be less variance in the sway variable in young adults because their physical function and fitness are at a higher level. However, study also showed young adults exhibited the same pattern of sway over the visual conditions as was observed among older adults, as revealed by measurements of the COP and kinematics of body segments [26]. Further investigation is necessary to better understand the exact impact of PA on static balance in different populations.

Additionally, the option of running shoes was chosen during the static tests in our study. Previous studies have shown significant differences in posture control have been found during walking or running [40,41]. One study [42] also reported significant difference existed in sway velocity between barefoot and wearing running shoes when standing on single foot. However, to our knowledge, little research has been done to examine the differences between shod and barefoot when standing with two feet on the force plate. In the future, we will conduct further tests to investigate the differences between barefoot and different footwear and how they affect the balance. 

This study had strengths as well as limitations. Strengths: First, we used the accelerometer to objectively measure PA, and we also conducted objective balance testing for static balance in this population. The results were both valid and reliable to reflect the association between PA and static balance. Second, most participants strictly followed our instructions of accelerometer wearing, and the adherence rate was good. The balance testing was conducted in a quiet and controlled environment to reduce the interference of noise on static balance. Limitations of this study included the following. (1) The individual differences in muscle strength may be related to static balance and were not considered in our study. Due to limitation of space and time, only static balance was measured, and dynamic balance was not tested in this study. Additionally, footwear may have extra support during the balance tests. (2) This study was a cross-sectional study, and the sample size of this study was relatively small. The participants were relatively young. (3) Total device wearing time was significantly higher in women. To avoid the influence of the variation in device wearing time, it was controlled as a covariate in the regression models. Intervention studies are needed to find out the causal relationship of adult’s static balance with their PA. Future studies with a larger sample size, different age groups, dynamic balance, and longitudinal measurements of fall risks are recommended.

## 5. Conclusions

This study suggests that static balance had significant associations with sex and visual contribution. Women, or those in the condition of eyes open, performed better in static balance. Higher MVPA and less SED were associated with lower SA, respectively, in young adults. Increasing PA levels and reduced sedentary behaviors may promote static balance, but further intervention studies are needed to better understand the effect of PA on static balance in this population.

## Figures and Tables

**Figure 1 ijerph-18-10787-f001:**
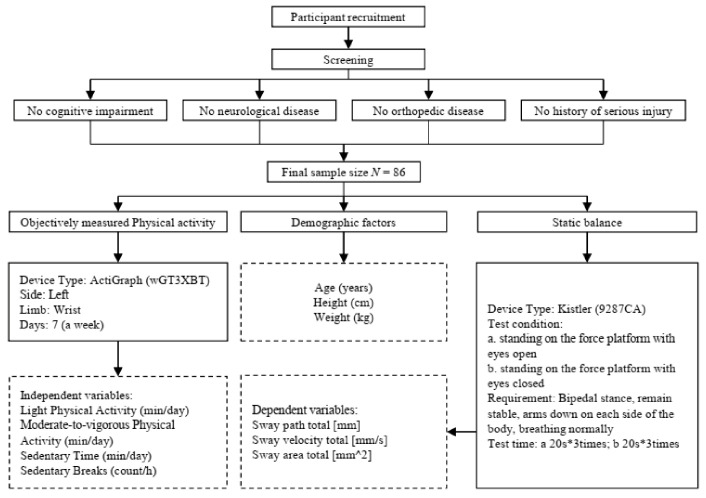
Flow chart of the study.

**Figure 2 ijerph-18-10787-f002:**
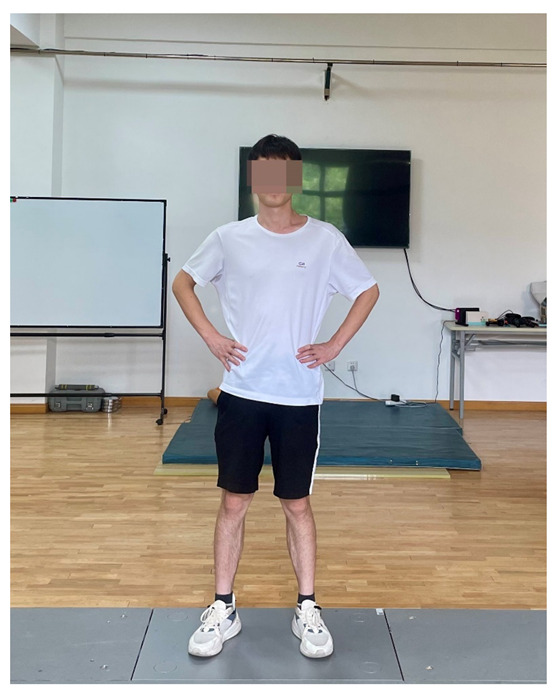
Posture during static balance test.

**Figure 3 ijerph-18-10787-f003:**
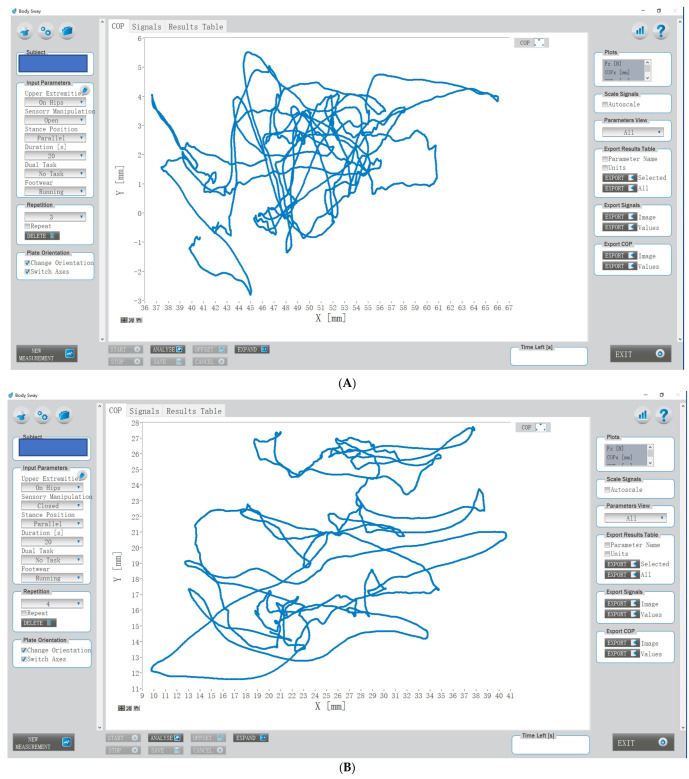
The movement track of the center of pressure (COP) of one participant during static balance tests. (**A**) eyes open; (**B**) eyes closed.

**Table 1 ijerph-18-10787-t001:** Characteristics of study population (Mean ± SD).

		Men	Women	Total
**Demographic Factors**	n	43	43	86
	Age (year)	21.93 ± 1.94	20.58 ± 2.07 **	21.26 ± 2.11
	Height (cm)	179.40 ± 7.79	166.51 ± 5.78 **	172.95 ± 9.41
	Weight (kg)	74.10 ± 12.58	60.46 ± 9.27 **	67.28 ± 12.95
	BMI (kg/m^2^)	22.97 ± 3.22	21.74 ± 2.60	22.36 ± 2.97
**Physical** **Activity Variables**	Total device wearing time (min/day)	660.37 ± 170.00	787.97 ± 218.03 **	724.17 ± 204.66
	SED (min/day)	365.47 ± 114.22	479.24 ± 189.18 **	422.35 ± 165.54
	LPA (min/day)	163.50 ± 46.91	190.35 ± 45.94 **	176.93 ± 48.09
	MVPA (min/day)	131.40 ± 67.05	118.38 ± 34.29	124.89 ± 53.34
	SB (count/h)	2.22 ± 2.10	3.89 ± 3.42 **	3.05 ± 2.94

** *p* < 0.01 indicates significant difference between different sex groups. Abbreviations: BMI = body mass index; LPA = light physical activity; MVPA = moderate-to-vigorous physical activity; SB = sedentary breaks; and SED = sedentary time.

**Table 2 ijerph-18-10787-t002:** Descriptive statistic of sway param (Mean ± SD).

Body Sway Variables	Eyes Open	Eyes Closed
**Men**		
SP-total [mm]	227.03 ± 44.76 *	276.43 ± 58.37 *^,#^
SV-total [mm/s]	11.35 ± 2.23 *	13.82 ± 2.92 *^,#^
SA-total [mm^2^]	264.28 ± 124.58	354.78 ± 143.18 *^,#^
**Women**		
SP-total [mm]	200.57 ± 41.33	229.41 ± 60.46 ^#^
SV-total [mm/s]	10.03 ± 2.07	11.47 ± 3.03 ^#^
SA-total [mm^2^]	229.28 ± 93.65	281.45 ± 165.98 ^#^
**Total**		
SP-total [mm]	213.80 ± 44.84	252.92 ± 63.63
SV-total [mm/s]	10.93 ± 2.62	12.67 ± 3.29
SA-total [mm^2^]	246.78 ± 110.96	318.12 ± 158.44

* *p* < 0.05 indicates significant difference between different sex groups. ^#^ *p* < 0.05 indicates significant difference between eyes open and eyes-closed condition. Abbreviations: SA = sway area; SP = sway path length; and SV = sway velocity.

**Table 3 ijerph-18-10787-t003:** Association ^a^ between physical activity and static balance.

	SP			SV			SA		
	Beta Estimate	95% C.I.	*p*	Beta Estimate	95% C.I.	*p*	Beta Estimate	95% C.I.	*p*
MVPA	−0.028	−0.284–0.228	0.829	−0.001	−0.014–0.011	0.833	−0.720	−1.370–0.071	0.030
LPA	−0.020	−0.303–0.263	0.888	−0.001	−0.015–0.013	0.889	0.137	−0.591–0.865	0.712
SED	−0.005	−0.138–0.128	0.943	−0.0003	−0.007–0.006	0.941	0.201	0.091–0.311	0.0004
SB	−1.917	−6.180–2.345	0.376	−0.097	−0.310–0.117	0.372	0.244	−10.753–11.240	0.965

^a^ Means the models were controlled for condition (eyes open/closed), sex, age, body mass index (BMI), total device wearing time, and interaction between physical activity variables and condition. Abbreviations: LPA means light physical activity; MVPA means moderate-to-vigorous physical activity; SB means sedentary breaks; SED means sedentary time; and SP-total = total sway path length.

## Data Availability

The data presented in this study are available on request from the corresponding author. The data are not publicly available due to them being a part of a doctoral study and additional articles are still being prepared.

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
