# Peer review of "Objectively Measured Physical Activity Is Associated with Static Balance in Young Adults"

_ijerph, 2021, doi:10.3390/ijerph182010787_

Round 1

Reviewer 1 Report

Comments to authors

Abstract

Line 13 – a space is needed before + when describing your SD. This error is common throughout the paper. Format according as an example  139 + .45

Line 14 – please list the brand of accerlerometer used and where they were placed on the body

Line 16-17 – What was the type of force plate used? This should be mention/listed in the abstract

Line 18-22 – a space is needed when stating your p values. p = . 04   Also, precise p values should be stated instead of showing p < .05  what was the precise p value?  Precise statistical methods and program should also be listed in the abstract. This is missing.

Introduction

Line 42 – Re-word the term, “free-living”. This is not understood and will need a description or citation.

Line 44 – sentence is incorrect. Insert “the” before the ‘older population’.

Methods

Line 76-77 – As stated earlier in the abstract, please format + with spacing when reporting SD

Justify your methodology of having subjects wearing shoes on the Kistler forceplate. This technique is not used due to the lack of validity of CoP and LoS scores to determine improvements when shoes provide additional support. Measurements are not considered true readings if not barefoot. Why did you use this technique? Using the foot shows validity on CoP studies, especially with static balance scores. I believe this methodology is flawed.

Justify your methodology of having subjects wearing the accelerometers on the wrist vs on the hip. Too many false activity readings are based on accidental movements of the limbs when this would not occur if the device was placed on the hip. If you believe your physical activity measurements were accurate, justify this as well with other published studies showing valid readings using the wrist site.

Justify the use of a linear regression over a Pearson Correlation to show a “relationship” between static balance scores and physical activity. Linear regressions predict more so, even though they are similar.

Comments to authors

Abstract

Line 13 – a space is needed before + when describing your SD. This error is common throughout the paper. Format according as an example  139 + .45

Line 14 – please list the brand of accerlerometer used and where they were placed on the body

Line 16-17 – What was the type of force plate used? This should be mention/listed in the abstract

Line 18-22 – a space is needed when stating your p values. p = . 04   Also, precise p values should be stated instead of showing p < .05  what was the precise p value?  Precise statistical methods and program should also be listed in the abstract. This is missing.

Introduction

Line 42 – Re-word the term, “free-living”. This is not understood and will need a description or citation.

Line 44 – sentence is incorrect. Insert “the” before the ‘older population’.

Methods

Line 76-77 – As stated earlier in the abstract, please format + with spacing when reporting SD

Justify your methodology of having subjects wearing shoes on the Kistler forceplate. This technique is not used due to the lack of validity of CoP and LoS scores to determine improvements when shoes provide additional support. Measurements are not considered true readings if not barefoot. Why did you use this technique? Using the foot shows validity on CoP studies, especially with static balance scores. I believe this methodology is flawed.

Justify your methodology of having subjects wearing the accelerometers on the wrist vs on the hip. Too many false activity readings are based on accidental movements of the limbs when this would not occur if the device was placed on the hip. If you believe your physical activity measurements were accurate, justify this as well with other published studies showing valid readings using the wrist site.

Justify the use of a linear regression over a Pearson Correlation to show a “relationship” between static balance scores and physical activity. Linear regressions predict more so, even though they are similar.

Comments to authors

Abstract

Line 13 – a space is needed before + when describing your SD. This error is common throughout the paper. Format according as an example  139 + .45

Line 14 – please list the brand of accerlerometer used and where they were placed on the body

Line 16-17 – What was the type of force plate used? This should be mention/listed in the abstract

Line 18-22 – a space is needed when stating your p values. p = . 04   Also, precise p values should be stated instead of showing p < .05  what was the precise p value?  Precise statistical methods and program should also be listed in the abstract. This is missing.

Introduction

Line 42 – Re-word the term, “free-living”. This is not understood and will need a description or citation.

Line 44 – sentence is incorrect. Insert “the” before the ‘older population’.

Methods

Line 76-77 – As stated earlier in the abstract, please format + with spacing when reporting SD

Justify your methodology of having subjects wearing shoes on the Kistler forceplate. This technique is not used due to the lack of validity of CoP and LoS scores to determine improvements when shoes provide additional support. Measurements are not considered true readings if not barefoot. Why did you use this technique? Using the foot shows validity on CoP studies, especially with static balance scores. I believe this methodology is flawed.

Justify your methodology of having subjects wearing the accelerometers on the wrist vs on the hip. Too many false activity readings are based on accidental movements of the limbs when this would not occur if the device was placed on the hip. If you believe your physical activity measurements were accurate, justify this as well with other published studies showing valid readings using the wrist site.

Justify the use of a linear regression over a Pearson Correlation to show a “relationship” between static balance scores and physical activity. Linear regressions predict more so, even though they are similar.

Comments to authors

Abstract

Line 13 – a space is needed before + when describing your SD. This error is common throughout the paper. Format according as an example  139 + .45

Line 14 – please list the brand of accerlerometer used and where they were placed on the body

Line 16-17 – What was the type of force plate used? This should be mention/listed in the abstract

Line 18-22 – a space is needed when stating your p values. p = . 04   Also, precise p values should be stated instead of showing p < .05  what was the precise p value?  Precise statistical methods and program should also be listed in the abstract. This is missing.

Introduction

Line 42 – Re-word the term, “free-living”. This is not understood and will need a description or citation.

Line 44 – sentence is incorrect. Insert “the” before the ‘older population’.

Methods

Line 76-77 – As stated earlier in the abstract, please format + with spacing when reporting SD

Justify your methodology of having subjects wearing shoes on the Kistler forceplate. This technique is not used due to the lack of validity of CoP and LoS scores to determine improvements when shoes provide additional support. Measurements are not considered true readings if not barefoot. Why did you use this technique? Using the foot shows validity on CoP studies, especially with static balance scores. I believe this methodology is flawed.

Justify your methodology of having subjects wearing the accelerometers on the wrist vs on the hip. Too many false activity readings are based on accidental movements of the limbs when this would not occur if the device was placed on the hip. If you believe your physical activity measurements were accurate, justify this as well with other published studies showing valid readings using the wrist site.

Justify the use of a linear regression over a Pearson Correlation to show a “relationship” between static balance scores and physical activity. Linear regressions predict more so, even though they are similar.

Author Response

Response to Reviewer 1 Comments

Point 1: Abstract: Line 13 – a space is needed before + when describing your SD. This error is common throughout the paper. Format according as an example 139 + .45

Response 1: Thanks for the comments. The space issue has been modified throughout the paper.

Point 2: Line 14 – please list the brand of accelerometer used and where they were placed on the body.

Response 2: The brand of accelerometer used and where they were placed on the body have been added in the abstract (Page 1, Line 14-15).

Point 3: Line 16-17 – What was the type of force plate used? This should be mentioned/listed in the abstract.

Response 3: The type of the force plate has been listed in the abstract (Page 1, Line 18).

Point 4: Line 18-22 – a space is needed when stating your p values. p = . 04   Also, precise p values should be stated instead of showing p < .05  what was the precise p value?  Precise statistical methods and program should also be listed in the abstract. This is missing.

Response 4: The spaces have been added in the text. The precise p value and the statistical methods have also been stated in the abstract. (Page 1, Line 18-24).

Point 5: Introduction: Line 42 – Re-word the term, “free-living”. This is not understood and will need a description or citation.

Response 5: The term has been modified (Page 2, Line 50).

Point 6: Line 44 – sentence is incorrect. Insert “the” before the ‘older population’.

Response 6: The word “the” has been inserted in the text (Page 2, Line 52).

Point 7: Methods: Line 76-77 – As stated earlier in the abstract, please format + with spacing when reporting SD

Response 7: Spaces have been added in the whole paper.

Point 8: Justify your methodology of having subjects wearing shoes on the Kistler force plate. This technique is not used due to the lack of validity of CoP and LoS scores to determine improvements when shoes provide additional support. Measurements are not considered true readings if not barefoot. Why did you use this technique? Using the foot shows validity on CoP studies, especially with static balance scores. I believe this methodology is flawed.

Response 8: Thanks for the good point. However, we believe this methodology is right and there are some previous studies using similar design with participants wearing running shoes [22,23]. Running shoes were used to test the posture control ability in real life setting. Participants in our study were required to wear standard running shoes and remain the bipedal standing posture on the platform (Figure 2). The movement track of the center of pressure (COP) was between their two legs, and the effect of shoes was less than one foot. In our study, the “Measurement, Analysis and Reporting Software” (MARS) was applied for comprehensive analyses of force plate measurements. There were three options in the software for footwear, including barefoot, running shoes and high heels, and the option of “running shoes” has been chosen during our tests. To clarify this point, more information has been provided in the Methods (Page 4, Line 134-139, Line 145-148).

Point 9: Justify your methodology of having subjects wearing the accelerometers on the wrist vs on the hip. Too many false activity readings are based on accidental movements of the limbs when this would not occur if the device was placed on the hip. If you believe your physical activity measurements were accurate, justify this as well with other published studies showing valid readings using the wrist site.

Response 9: Thanks for the comments. Large observational studies, such as the National Health and Nutrition Examination Survey (NHANES) study have changed their protocols from attachment on hips to wrists [17]. Some studies reported reasonable precise estimations of PA when using wrist-worn devices [17, 18]. To some extent wear methods are dependent on the study aim, the design of the accelerometers, as well as acceptability within the study population. In our pilot study, participants preferred to wear the accelerometer on the wrist because it more comfortable, convenient, and easy to remember. To improve the adherence of the device wearing, the attachment on wrists was chosen. This point has been added in the text (Page 3, Line 112-114).

Point 10: Justify the use of a linear regression over a Pearson Correlation to show a “relationship” between static balance scores and physical activity. Linear regressions predict more so, even though they are similar.

Response 10: Thanks for the information. The linear regression was used because those confounders can be properly controlled in the regression model, and it is more commonly to be used in previous studies.

Reviewer 2 Report

Major revisions are needed of the statistical analysis. Regarding the main research question, association between balance variables (sway parameters) and physical activity variables are presented in Tables 3, 4 and 5. In total, 144 “B” parameters are presented in these tables. With a given alpha (0.05), 7 of these can be expected on average to be significant, just by random. Thus, as there are only 5 of these significant in the current analyses, it is highly likely that these significant numbers have no meaning at all. The accumulation of Type I statistical errors needs to be addressed and dealt with in the statistical analysis and the discussion of manuscript. There are several options to do that. One would be to use Bonferroni adjustments of p-values or the alpha value. Another, better option, would be to also test fewer models – by for example including sex and open/closed eyes as control variables in the models and/or combining, or excluding, some the sway parameters. For the effect statistics, it would be more informative to use standardize beta or partial r-square with confidence interval, instead of B and SE (Tables 3, 4 and 5).

Using t-tests to compare the sexes and open/closed eyes conditions in Tables 1 and 2 is questionable. First, differences between the sexes and the eye conditions are not among the main study questions. Actually, the effect of visual input is stated as a secondary aim, but the rational for that aim is missing. Secondly, it is more appropriate to control for these variables in the statistical models.

The study population is young adults, age 18-25 y.o. However, most of the cited literature is on older adults. It is well known that both physical activity and balance control is very different between this two age groups. Thus, the current results should not be directly compared to the literature on older adults, without some reservations. The author need to rewrite the introduction and discussion sections where they consider better the distinction between these two age groups, or consider only the literature on young adults.

Author Response

Response to Reviewer 2 Comments

Point 1: Major revisions are needed of the statistical analysis. Regarding the main research question, association between balance variables (sway parameters) and physical activity variables are presented in Tables 3, 4 and 5. In total, 144 “B” parameters are presented in these tables. With a given alpha (0.05), 7 of these can be expected on average to be significant, just by random. Thus, as there are only 5 of these significant in the current analyses, it is highly likely that these significant numbers have no meaning at all. The accumulation of Type I statistical errors needs to be addressed and dealt with in the statistical analysis and the discussion of manuscript. There are several options to do that. One would be to use Bonferroni adjustments of p-values or the alpha value. Another, better option, would be to also test fewer models – by for example including sex and open/closed eyes as control variables in the models and/or combining, or excluding, some the sway parameters. For the effect statistics, it would be more informative to use standardize beta or partial r-square with confidence interval, instead of B and SE (Tables 3, 4 and 5).

Response 1: Thanks for the good comments. The statistical analysis has been modified, and sex has been included as one of the control variables in the models. Sway velocity has been deleted because it did not provide any new information. For the effect statistics, the standardized beta and 95% confidence interval have been used instead of B and SE (Table 3, 4 and 5). The open/closed eyes conditions were still kept to classify the results because they were completely different measure conditions and need to be discussed separately.

Point 2: Using t-tests to compare the sexes and open/closed eyes conditions in Tables 1 and 2 is questionable. First, differences between the sexes and the eye conditions are not among the main study questions. Actually, the effect of visual input is stated as a secondary aim, but the rational for that aim is missing. Secondly, it is more appropriate to control for these variables in the statistical models.

Response 2: The comparisons between sexes and conditions were for description of the characteristics of the study population. Literature has shown sex was one of the important factors that influenced in postural balance [3]. Women showed higher postural stability in different conditions and measurements [4]. Visual inputs could also be crucial for postural control in healthy adults. Stability with the eyes closed were worse than those obtained with the eyes open [5]. It is necessary to describe the difference between varied sexes or conditions in our participants. We have clarified this point in the text (Page 1-2, Line 38-41).

Point 3: The study population is young adults, age 18-25 y.o. However, most of the cited literature is on older adults. It is well known that both physical activity and balance control is very different between this two age groups. Thus, the current results should not be directly compared to the literature on older adults, without some reservations. The author need to rewrite the introduction and discussion sections where they consider better the distinction between these two age groups, or consider only the literature on young adults.

Response 3: Thanks for the comments. Those studies related to older adults were important and related to our study, because they addressed the significant association between physical activity and balance in a specific population, indicating more effort should be done to investigate this area. Limited investigations have been found focusing on the young adults, but there were still some studies about young adults and their balance ability were cited in the manuscript [30-33, 37]. We have also added this point as one of our limitations in the Discussion (Page 11, Line 321). However, physical inactivity may also lead to poorer performance in balance and higher risk of fall at an earlier age. Investigation of the association between PA and static balance in younger adults who also develop an inactive lifestyle is warranted to understand how to optimize the prescription of PA strategies or interventions designed to improve posture stability in daily activities in general population.

Round 2

Reviewer 1 Report

I wished the authors would have added additional research on shod vs barefoot on the force plates with regards to CoP readings, however will accept as is.

Author Response

Response to Reviewer 1 Comments

Point 1: I wished the authors would have added additional research on shod vs barefoot on the force plates with regards to CoP readings, however will accept as is.

Response 1: Thanks for the comments. We admit that there are significantly differences in posture control during walking or running [40,41]. One study [42] also reported significant difference existed in sway velocity between barefoot and wearing running shoes when standing on single foot. However, to our knowledge, little research has been done to examine the differences between shod vs barefoot when standing with two feet on the force plate. In the future, we will conduct further tests to investigate the differences between barefoot and different footwear and how it affects the balance. This point has been added in the discussion (Page 10, Line 276-283).

Reviewer 2 Report

Reviewer´s comments on Response 1: There still are a lot of problems with the statistics. Now, there are 48 p-values in Tables 3 and 4, and on average 2 or 3 of them can be expected to be significant by random only. In the current manuscript only two of them are significant. Thus the claim by the authors of an effect must be considered very weak, and should be more clearly pointed out in limitations of the study. The explanation for this weak results could also be discussed. In Table 3 and 4, association between physical activity and static balance variables are tested. Even though eye conditions affect static balance (see also comment on Response 2), they may not necessarily affect the association between static balance and physical activity. This should be tested statistically (e.g. by testing interaction between the condition and physical activity variables and adjust for repeated measures), to justify and before the data is separated by eye conditions in Tables 3 and 4. Although the static balance parameters have been reduced from 9 to 6 (sway area/sway path and total/AP/ML), it is still unclear what different parameters represent and why different association can be expected for different parameters, i.e. why you need to present the results for all of them separately. This should be reasoned, or at least discussed. Preferably, you could test statistically if there are any differences between these parameters in regards to the association with the physical activity variables (by including interaction terms into the statistical models). The authors say in their Response 1: “The open/closed eyes conditions were still kept to classify the results because they were completely different measure conditions and need to be discussed separately” (reviewer’s underlining) – but this is not separately discussed in your Discussion section at all. And, actually should be tested statistically if it is going to be discussed. Then, in Tables 3 and 4, the 95% confidence intervals do not match the standardized beta (it seems the confidence intervals are unstandardized). Also, it is not clear enough when the upper confidence limits are negative and when they are positive values. Finally, there are some discrepancies between the confidence intervals and the p-values. For example, for the SP-AP versus Eyes Open/SED in Table 3, the standardized beta is 0.178, the C.L. -0.007-0.148, and the p-value is 0.532. I assume the C.L. should are –0.007;+0.148, but in any case, the standardized beta (+0.178) is outside the interval! Beside, this interval should mean much lower p-value.

Reviewer´s comments on Response 2:  As the authors state, both sex and eye conditions affect static balance and this is very well established in published papers. Thus yes, there is a good reason to present the descriptive statistic in Table 2, separately for men/women and eyes open/closed. However, it is not clear what the current study adds to the existing literature regarding this differences, and thus why the secondary aim of the paper is stated. Therefore, it does not justify the inferential testing of these differences in Table 2, or the secondary aim should be better reasoned. Furthermore, this cannot be considered as a reason to present the models in Tables 3 and 4, separately for eyes open/closed (see more on that in my comment on Response 1).

Reviewer´s comments on Response 3:  Thanks. I still think there can be more done to incorporate or stress this points in your response into the manuscript. For example, “young adults” could be included in both the title of the paper and objectives. Also, you could include the statement “Limited investigations have been found focusing on the young adults” in your introduction, etc. Furthermore, differences between young adults and older people could be addressed better in either the Introduction or the Discussion section.

Author Response

Response to Reviewer 2 Comments

Point 1  Reviewer´s comments on Response 1: There still are a lot of problems with the statistics. Now, there are 48 p-values in Tables 3 and 4, and on average 2 or 3 of them can be expected to be significant by random only. In the current manuscript only two of them are significant. Thus the claim by the authors of an effect must be considered very weak, and should be more clearly pointed out in limitations of the study. The explanation for this weak results could also be discussed. In Table 3 and 4, association between physical activity and static balance variables are tested. Even though eye conditions affect static balance (see also comment on Response 2), they may not necessarily affect the association between static balance and physical activity. This should be tested statistically (e.g. by testing interaction between the condition and physical activity variables and adjust for repeated measures), to justify and before the data is separated by eye conditions in Tables 3 and 4. Although the static balance parameters have been reduced from 9 to 6 (sway area/sway path and total/AP/ML), it is still unclear what different parameters represent and why different association can be expected for different parameters, i.e. why you need to present the results for all of them separately. This should be reasoned, or at least discussed. Preferably, you could test statistically if there are any differences between these parameters in regards to the association with the physical activity variables (by including interaction terms into the statistical models). The authors say in their Response 1: “The open/closed eyes conditions were still kept to classify the results because they were completely different measure conditions and need to be discussed separately” (reviewer’s underlining) – but this is not separately discussed in your Discussion section at all. And, actually should be tested statistically if it is going to be discussed. Then, in Tables 3 and 4, the 95% confidence intervals do not match the standardized beta (it seems the confidence intervals are unstandardized). Also, it is not clear enough when the upper confidence limits are negative and when they are positive values. Finally, there are some discrepancies between the confidence intervals and the p-values. For example, for the SP-AP versus Eyes Open/SED in Table 3, the standardized beta is 0.178, the C.L. -0.007-0.148, and the p-value is 0.532. I assume the C.L. should are –0.007;+0.148, but in any case, the standardized beta (+0.178) is outside the interval! Beside, this interval should mean much lower p-value.

Response 1: Thank you for the good suggestions. It is really helpful. New statistical analyses have been conducted to test the interaction between the condition and physical activity variables and adjust for repeated measures. The new results have been updated in Table 3. The association between PA and static balance under eye open/closed condition was analyzed using linear mixed-effects models. Fixed effects of PA variables (MVPA, LPA, SED or sedentary breaks), condition (eyes open/closed), age, sex, body mass index (BMI), and total device wearing time and the PA*condition interaction were included in the models, and random intercepts over time were included to account for within-subject correlations. We have also updated the methods in the text (Page 6, Line 194-200). The sway in anterior-posterior and medio-lateral directions have been deleted to help to illustrate the main point of the results. The associations between PA and static balance were presented in Table 3. We found MVPA was negatively associated with SA (p = 0.030), and SED were positively associated SA (p = 0.0004). No associations were found in other aspect (p > 0.05). We admit the association between physical activity and static balance was still not very strong. To make the conclusion more cautious, we have discussed the measures of sway path, sway velocity and sway area in the text (Page 11, Line 336-348). Also, the association was restricted only between MVPA/SED and sway area through the whole paper.

Point 2: Reviewer´s comments on Response 2:  As the authors state, both sex and eye conditions affect static balance and this is very well established in published papers. Thus yes, there is a good reason to present the descriptive statistic in Table 2, separately for men/women and eyes open/closed. However, it is not clear what the current study adds to the existing literature regarding this differences, and thus why the secondary aim of the paper is stated. Therefore, it does not justify the inferential testing of these differences in Table 2, or the secondary aim should be better reasoned. Furthermore, this cannot be considered as  reason to present the models in Tables 3 and 4, separately for eyes open/closed (see more on that in my comment on Response 1).

Response 2: Thanks for the comments. Table 3-4 have been merged to Table 3 and new results have been presented in the table. New analyses methods have been mentioned in Response 1. The secondary aim has been deleted to avoid confusion. The comparisons in Table 1-2 were to provide descriptive data for men and women under eyes open or closed condition, also give rational for adjusting sex and condition as covariates in the regression models in Table 3.

Point 3: Reviewer´s comments on Response 3:  Thanks. I still think there can be more done to incorporate or stress this points in your response into the manuscript. For example, “young adults” could be included in both the title of the paper and objectives. Also, you could include the statement “Limited investigations have been found focusing on the young adults” in your introduction, etc. Furthermore, differences between young adults and older people could be addressed better in either the Introduction or the Discussion section.

Response 3: Thanks for the advice. The term of “young adults” has been added in the title and the whole manuscript. the results of this study were mainly focusing on young adults. The differences between young and older adults have been explained more in the manuscript and several new references have been added (Page 11, Line 348-358). Research has indicated significant difference in balance and posture control in young and older adults. Better posture control ability and less sway existed in balance measures among young participants, compared to older adults [26,38-39]. There may be less variance in the sway variable in young adults because their physical function and fitness are at a higher level. Older adults had higher electrophysiological costs for a given task and so can be considered less efficient from a neuromuscular viewpoint [39]. However, study also showed the young adults exhibited the same pattern of sway over the visual conditions as was observed among older adults, as revealed by measurements of the COP and kinematics of body segments [26]. Further investigation was necessary to better understand the exact impact of PA on static balance in young and older adults.

This manuscript is a resubmission of an earlier submission. The following is a list of the peer review reports and author responses from that submission.

Round 1

Reviewer 1 Report

I send my suggestions in attached file

Reviewer 2 Report

Attached file
